# Cost-Effectiveness of Three Poliovirus Immunization Schedules in Shanghai, China

**DOI:** 10.3390/vaccines9101062

**Published:** 2021-09-23

**Authors:** Jia Ren, Hairenguli Maimaiti, Xiaodong Sun, Zhuoying Huang, Jiechen Liu, Jianping Yang, Zhi Li, Qingrui Bai, Yihan Lu

**Affiliations:** 1Shanghai Municipal Center for Disease Control and Prevention, Institute of Immunization, Shanghai 200336, China; renjia@scdc.sh.cn (J.R.); sunxiaodong@scdc.sh.cn (X.S.); huangzhuoying@scdc.sh.cn (Z.H.); liu-jiechen@scdc.sh.cn (J.L.); yangjianping@scdc.sh.cn (J.Y.); lizhi@scdc.sh.cn (Z.L.); baiqingrui@scdc.sh.cn (Q.B.); 2Ministry of Education Key Laboratory of Public Health Safety, Department of Epidemiology, School of Public Health, Fudan University, Shanghai 200032, China; rglmmthai20@fudan.edu.cn

**Keywords:** poliomyelitis, poliovirus, live attenuated oral polio vaccine, inactivated polio vaccine, cost-effectiveness analysis, cost-benefit analysis

## Abstract

In Shanghai, China, a polio immunization schedule of four inactivated polio vaccines (IPV) has been implemented since 2020, replacing the schedules of a combination of two IPVs and two bivalent live attenuated oral polio vaccines (bOPV), and four trivalent live attenuated oral polio vaccines (tOPV). This study aimed to assess the cost-effectiveness of these three schedules in infants born in 2016, in preventing vaccine-associated paralytic poliomyelitis (VAPP). We performed a decision tree model and estimated incremental cost-effectiveness ratio (ICER). Compared to the four-tOPV schedule, the two-IPV-two-bOPV schedule averted 1.2 VAPP cases and 16.83 disability-adjusted life years (DALY) annually; while the four-IPV schedule averted 1.35 VAPP cases and 18.96 DALY annually. Consequently, ICER_VAPP_ and ICER_DALY_ were substantially high for two-IPV-two-bOPV (CNY 12.96 million and 0.93 million), and four-IPV (CNY 21.24 million and 1.52 million). Moreover, net monetary benefit of the two-IPV-two-bOPV and four-IPV schedules was highest when the cost of IPV was hypothesized to be less than CNY 23.75 or CNY 9.11, respectively, and willingness-to-pay was hypothesized as CNY 0.6 million in averting one VAPP-induced DALY. IPV-containing schedules are currently cost-ineffective in Shanghai. They may be cost-effective by reducing the prices of IPV, which may accelerate polio eradication in Chinese settings.

## 1. Introduction

Poliomyelitis is an acute intestinal infection caused by poliovirus (PV), which may result in lifelong disability in children. Large-scale global use of the polio vaccine has dramatically reduced 99% of polio incidence [1,2]. The latest local poliomyelitis case of wild poliovirus (WPV) was notified in 1987 in Shanghai, and the latest indigenous case was reported in 1994 in China [3,4]. In 2000, China was certified as free of polio [2,5]. After containing the imported cases and subsequent epidemic of poliomyelitis in northwestern China in 2011, China was affirmed as having polio-free status in 2012 [2,3]. Currently, high polio vaccination coverage and continual surveillance of acute flaccid paralysis (AFP) cases are maintained in China [2]. However, China remains at risk of WPV due to bordering on polio-endemic countries, namely, Pakistan and Afghanistan [2,6].

The use of live attenuated oral polio vaccine (OPV) in China since 1965 has resulted in a significant decrease in the incidence of poliomyelitis [7]. However, OPV may cause vaccine-associated paralytic poliomyelitis (VAPP) in rare cases, and its clinical symptoms are difficult to distinguish from those caused by WPV [8]. The incidence of VAPP cases in Shanghai was 3.01 cases per million for the first dose of OPV in 1980–1998 [4]. In addition, mutation in the vaccine strain virus of OPV may induce vaccine-derived poliovirus cases (VDPVs), due to long-term circulation in humans or the environment [8]. In 2016, the World Health Organization (WHO) recommended a global switch in immunization strategy from trivalent OPV (tOPV) (containing serotype 1, 2 and 3) to the bivalent version (bOPV) (containing serotype 1 and 3) in the national immunization program, and inclusion of at least one dose of inactivated poliovirus vaccine (IPV) in routine immunization schedules [1]. China subsequently implemented the combined immunization schedule of one IPV and three bOPVs that year [3]. In the same year, Shanghai was the first provincial administrative region to adopt the schedule of two IPVs and two bOPVs in China [9,10]. Furthermore, Shanghai has adopted a four-IPV schedule since October 2020 [11].

However, the economic benefits of IPV-containing immunization schedules remained unclear, which might delay its expanding implementation worldwide. Findings from the health economics evaluation of replacing OPV with IPV varied widely, such as incremental cost-effectiveness ratio (ICER) for preventing one VAPP case ranging between USD 0.74 and 80.0 million and ICER for averting one corresponding disability-adjusted life year (DALY) caused by VAPP ranging between USD 0.009 and 5.6 million [12,13,14,15,16,17,18,19,20]. In addition, IPV inclusion in the immunization schedule remains cost-ineffective, though it differed extensively across the studies [17,18,21,22,23,24,25]. This study is aimed to determine and compare the cost-effectiveness and cost-benefit of averting VAPP by three polio immunization schedules in Shanghai, which would provide convincing evidence to further switch polio immunization strategy in China.

## 2. Materials and Methods

We simulated the utilization of three polio immunization schedules, including four-tOPV (vaccinated at 2, 3, and 4 months, and 4 years of age), two-IPV-two-bOPV (2, 3, and 4 months, and 4 years of age), and four-IPV (2, 3, 4, and 18 months of age), in a birth cohort of 2016 in Shanghai. The two-IPV-two-bOPV schedule replaced the traditional four-tOPV schedule in May 2016; furthermore, the four-IPV schedule has replaced the two-IPV-two-bOPV schedule since October 2020. Thus, during this study, the four-IPV schedule is the single polio immunization program implemented in Shanghai. A decision tree model was designed to estimate total cost of immunization, averted VAPP cases and DALYs, and then calculate corresponding ICER, for three polio immunization schedules in a cost-effectiveness analysis. Moreover, we hypothesized a willingness to pay (WTP) and determined net monetary benefit (NMB) for three immunization schedules, in a cost-benefit analysis. This study used routinely collected data from literature and the Shanghai Municipal Center for Disease Control and Prevention (SCDC) and involved no personal data, so it is exempted from ethical approval.

### 2.1. Definitions of AFP and VAPP

AFP cases were those who presented with symptoms of acute flaccid paralysis at less than 15 years of age or with a clinical diagnosis of polio at any age [26]. VAPP cases were described elsewhere [27]. Furthermore, VAPP cases were classified into recipient VAPP and contact VAPP according to OPV immunization history [28]. A recipient VAPP was defined as a case with fever occurring 4–35 days after vaccination, AFP occurring 6–40 days after vaccination, and a definite clinical diagnosis. A contact VAPP was a case having close contact with OPV-immunized persons within 35 days after vaccination and developed AFP 6–60 days after exposure, or with no OPV immunization 40 days prior to onset of paralysis and a definite clinical diagnosis. Both recipient and contact VAPP are confirmed by isolation of vaccine-related poliovirus from feces, pharynx, cerebrospinal fluid, brain, or spinal cord with very limited nucleotide mutations within VP1 sequence, compared to the Sabin strain [26]. 

### 2.2. Construction of a Decision Tree Model

A cohort of 218,400 infants born and routinely immunized between 1 January and 31 December 2016, in Shanghai, was used as a birth cohort in this study [29]. A decision tree model was constructed by using TreeAge Pro 2011 (TreeAge Software Inc., Williamstown, MA, USA). In the model, we hypothesized two states, being healthy and diagnosed as VAPP, for the vaccinated children. In addition, the model had the following hypotheses: (1) 100% coverage of polio immunization, (2) no self-paid polio-containing vaccines, (3) no occurrence of VAPP after four-IPV immunization in the study population; and (4) cost of cold-chain services, human resources, and administration (including training on the implementation of new immunization schedules, supervision, and social mobilization) remain identical across three polio immunization schedules (Figure 1).

Subsequently, total annual cost and corresponding sums of life years (LY) were estimated based on the probability of being healthy and VAPP in the context of different polio immunization schedules. Two IPV-containing schedules were compared to the four-tOPV schedule, in terms of additional cost per VAPP case prevented and corresponding ICER. In addition, univariate sensitivity analysis was performed for key variables in the model, using Tornado plot as described elsewhere [30,31,32].

### 2.3. Measurement of Cost

In this study, the cost was defined as being associated with the implementation of different polio immunization schedules, including procurement of vaccines, storage of vaccines, and compensation for VAPP cases.

#### 2.3.1. Procurement of Vaccines

First, polio vaccine, as an Expanded Programme on Immunization (EPI) vaccine, is purchased by SCDC. In this study, four categories of polio vaccines were involved, which had government procurement prices in 2016 as Chinese Yuan (CNY) 9.7/vial (for 10-dose) for tOPV, CNY 27.2/vial (for 10-dose) for bOPV, CNY 35/vial (for 1-dose) for Sabin IPV (sIPV), and CNY 39/vial (for 1-dose) for wild-type IPV (wIPV).

Second, we included two categories of IPV (sIPV and wIPV) to meet the demands of the IPV-containing immunization schedule in this study, as there were only two vaccine manufacturers in China that could supply IPV in 2016 and their production capacities were limited. SCDC has previously procured sIPV and wIPV at a ratio of 2:1–1:1, the baseline value was determined at 2:1.

Third, tOPV and bOPV are multi-dose vial vaccines, which may induce loss of doses in use. According to the Practical Regulations for Vaccination in Shanghai (2017 edition) [33] and SCDC data [34], the baseline coefficient of loss for OPV was determined to be 1.8 (range, 1.8–2.5), and that of IPV was 1 (range, 1–1.05) in this study.

Therefore, we calculated the cost per OPV dose = procurement price per OPV dose × coefficient of loss. Moreover, cost per IPV dose = procurement price per sIPV dose × procurement ratio of sIPV and wIPV/(procurement ratio of sIPV and wIPV + 1) + procurement price per wIPV dose × 1/(procurement ratio of sIPV and wIPV + 1) × coefficient of loss.

#### 2.3.2. Storage of Vaccines

OPV and IPV have different requirements of storage, which are attributable to the cold-chain equipment, including −20 °C for OPV (low-temperature refrigerator) and 2–8 °C for IPV (ordinary medical refrigerator). We defined that cost of low-temperature refrigerator = procurement price × number of vaccination clinics/years of depreciation. The depreciation rate of cold-chain equipment is 8–10 years, and the procurement price of a low-temperature refrigerator is about CNY 10,000–20,000 (baseline value was CNY 15,000). It was hypothesized that one low-temperature refrigerator is required for each community vaccination clinic (*n* = 265 in Shanghai Municipality).

#### 2.3.3. Compensation for VAPP Cases

Financial compensation for VAPP by Shanghai Municipal Government and its standards were specified in the Shanghai Compensation Measures for Vaccination Related Adverse Events (trial edition) in 2015 [35]. The amount of compensation would not exceed the maximum standards in each damage level. A total of five recipient VAPP cases due to tOPV vaccination in 2011–2014 were compensated, whereas one contact VAPP did not. Of them, 4 cases had damage levels of Grade II D and 1 case had Grade II B. With the cost of humanitarian compensation excluded, possible compensation for recipient VAPP cases in 2016 was calculated based on the maximum standards of compensation for Grade II D and Grade II B, which were CNY 748,005 and CNY 997,340, respectively. Considering that VAPP cases of Grade II D were predominant (80%), the baseline value of compensation for VAPP was set as CNY 748,005 in this study. Furthermore, a Chinese study in 2014 reported economic burden for VAPP cases with a disabling illness was CNY 1.18 million [21], which was discounted to compensation for VAPP cases in 2016 with a discount rate of 3%.

#### 2.3.4. Total Cost

Total cost was the sum of the above three parts, as cost of cold-chain services, human resources, and administration were hypothesized to be identical across the three polio immunization schedules. A formula was designed as follows:Total cost of polio immunization = C_V_ × D × A + C_M_ + B_i_ × A × E,(1)
where, A, number of births; B_i_, compensation for one VAPP case by implementing different immunization schedules; C_M_, maintenance of low-temperature refrigerators; C_V_, procurement of vaccines; D, administered doses; and E, the incidence of VAPP cases.

### 2.4. Measurement of Effectiveness

In this study, we presented number of VAPP cases prevented and related DALY averted as effectiveness.

#### 2.4.1. Incidence of VAPP Cases

The incidence of VAPP cases in the context of three polio immunization schedules was estimated based on the combination of local VAPP cases notified in Shanghai and literature review (Appendix A). We have a hypothesis that there would be no VAPP case in the context of the four-IPV immunization schedule.

#### 2.4.2. DALY Caused by VAPP Cases

It has previously been documented that there were 13 DALYs lost per VAPP case in low-income countries, while 14 DALYs in lower-middle, upper-middle, and high-income countries [12,14]. China has been classified as an upper-middle-income country [36], so baseline value was determined to be 14 DALYs per VAPP case (range, 13–14) in the model. In addition, according to the Shanghai Statistical Yearbook 2017 [29], life expectancy of local residents in Shanghai was 83.18 years in 2016.

### 2.5. Measurement of ICER

ICER is the ratio of incremental cost to incremental effectiveness [37]. In this study, incremental cost was the difference in the cost, and incremental effectiveness was the difference in preventing VAPP and averting DALYs, between IPV-containing schedules and the four-tOPV schedule. Low ICER means low incremental cost of preventing VAPP, using IPV-containing schedules compared to the four-tOPV schedule, which may have larger public health significance in polio immunization strategy.

Moreover, vaccines with an ICER ≤ 3 times of gross domestic product (GDP) are classified as cost-effective, while those with an ICER > 3 times that of GDP are cost-ineffective by the WHO guidelines [38]. The per capita GDP in Shanghai was CNY 116,562 in 2016.

### 2.6. Determination of WTP

WTP towards averting one VAPP case varied widely across the studies [21,22]. In a retrospective study in 2006, it was approximately USD 890,000 which meant a WTP of USD 64,000 towards averting one DALY caused by VAPP [14]. Considering a 3% discount rate and exchange rate, this was equal to approximately CNY 600,000, which was input into the decision tree model. In addition, we defined that net monetary benefit (NMB) = total benefit – total cost, in which total benefit = effectiveness (sum of LY per year) × bound of ICER_LY_ (that is WTP) and total cost was the total annual cost of different polio immunization schedules. Effect of possible variables on the NMB of the polio immunization schedules was presented in one-way sensitivity analysis net benefits graph, by using TreeAge Pro 2011 (TreeAge Software Inc., Williamstown, MA, USA).

## 3. Results

### 3.1. Cost-Effectiveness across Three Polio Immunization Schedules

Parameters as inputs in the decision tree model are presented in Table 1. We estimated the cost of tOPV (CNY 1.75–2.43 per dose), bOPV (CNY 4.90–6.80 per dose), and IPV (CNY 36.33–38.85 per dose). The cost of maintenance for all the low-temperature refrigerators in Shanghai was estimated to be between CNY 265,000 and 662,500 per year for OPV. In addition, compensation cost for VAPP was determined to be CNY 1,250,000 in 2016 with a range of CNY 748,005–1,250,000. Therefore, the total annual cost of polio immunization schedules was calculated to be CNY 2.98 million (four-tOPV), 18.56 million (two-IPV-two-bOPV), and 31.74 million (four-IPV) (Table 2).

Moreover, we estimated the effectiveness. Based on the incidence of VAPP and number of local resident births in 2016 (Table 1), the four-tOPV immunization schedule resulted in 1.35 VAPP cases per year, equivalent to 18.96 DALYs. In contrast, IPV-containing schedules averted corresponding VAPP cases and losses of DALYs, though they had additional costs (Table 2). ICER_DALY_ for the two-IPV-two-bOPV schedule and the four-IPV schedule was CNY 0.93 million and CNY 1.52 million, respectively, suggesting they were cost-ineffective (Table 2, Figure 2).

### 3.2. Sensitivity Analysis of Cost-Effectiveness

The incidence of VAPP cases with the four-tOPV schedule and the two-IPV-two-bOPV schedule, compensation cost of VAPP cases, and cost of IPV were included in the univariate sensitivity analysis according to the range of values (Table 1) to evaluate the influence of changes in these parameters on ICER_LY_. The change in the incidence of VAPP cases with four-tOPV had the largest effect on ICER_LY_, in the context of both the two-IPV-two-bOPV and four-IPV schedules (Figure 3a,b), which was further determined to be a negative association. In addition, ICER_LY_ was associated with cost of IPV (Figure 3a,b) and incidence of VAPP cases with the two-IPV-two-bOPV schedule (Figure 3a), which were both positive associations.

### 3.3. Cost-Benefit across Three Polio Immunization Schedules

We further determined the effects of WTP towards averting one DALY caused by VAPP on the decision tree model. When IPV cost < CNY 23.75, the two-IPV-two-bOPV schedule had a higher NMB, compared to the four-tOPV schedule; furthermore, when IPV cost < CNY 9.11, the four-IPV schedule had the highest NMB, (Figure 4a). In addition, when the incidence of VAPP cases with the four-tOPV schedule was >8.95 cases per million births, the two-IPV-two-bOPV schedule had a higher NMB, compared to the four-tOPV and the four-IPV schedules (Figure 4b).

## 4. Discussion

In this study, we found that the two-IPV-two-bOPV and the four-IPV schedules could prevent 1.20–1.35 VAPP cases and avert 16.83–18.96 DALYs per year, compared to the previous four-tOPV schedule in Shanghai, China. Similar studies have been previously reported. In Russia, incidence of VAPP cases was much lower for sequential immunization schedule of OPV and IPV (1 case/4.18 million doses) than that for the OPV schedule (1 case/1.59 million OPV doses) [41]. In the United States, no VAPP occurred for the IPV-containing schedule, compared to the incidence of VAPP being 2.9 cases/1 million OPV doses [42]. However, the two-IPV-two-bOPV and the four-IPV schedules had additional costs. ICER for preventing one VAPP and averting DALY caused by VAPP varied widely [12,13,14,15,16,17,18,19]. In this study, ICER_VAPP_ was between approximately USD 2.01 million and 3.29 million, while ICER_DALY_ was between approximately USD 0.14 million and 0.23 million. In 1996, an American study of children <6 years of age reported an ICER_VAPP_ of USD 3.1 million per VAPP for the two-IPV-two-bOPV schedule and USD 3 million per VAPP for the four-IPV schedule [19]. In contrast, an Australian study in 2001 of children <6 years showed that the ICER_VAPP_ was USD 10 million per VAPP and USD 29 million per VAPP, for these two schedules [18]. In addition, ICER_DALY_ in Columbia was USD 0.071 million per DALY [15]. The disparity in ICER_VAPP_ across countries may be related to the differing cost between IPV and OPV, incidence of VAPP, and compensation cost for VAPP in different countries.

However, the two-IPV-two-bOPV and the four-IPV immunization schedules were cost-ineffective, according to the WHO guidelines [38]. Sensitivity analysis of ICER showed that the incidence of VAPP with the four-tOPV schedule and cost of IPV had a larger impact on ICER (Figure 3). Furthermore, we found that when the cost of IPV decreased to CNY 23.75 or 9.11 per dose or the incidence of VAPP with the four-tOPV schedule exceeded 8.95 cases per million births, IPV-containing schedules had higher benefits, as shown in Figure 4, under a hypothetical WTP of CNY 600,000 in the decision tree. Our findings suggested that lower cost of IPV and higher VAPP incidence may contribute to lower ICER and higher cost-effectiveness. However, ICER varies across countries with different economic status [15,17,18,19,21]. In the upper-middle-income countries, polio immunization schedules save money in addition to preventing disease, which means the expected cost of treatment exceeds the difference in the cost between the schedules. In contrast, in low-income countries, the cost of immunization schedules remains relatively higher and the treatment cost is much lower, which implies a need to pay additional cost to obtain effectiveness [12]. It suggested possible limitations of utilizing ICER in the vaccine evaluation. Fundamental issues for deciding on the introduction of a vaccine may include the disease, the vaccines, and the strength of immunization program and health system [43].

The three polio immunization schedules in this study have both advantages and disadvantages in terms of safety, effectiveness, accessibility, and cost of administration [14,19]. In order to achieve the goal of polio eradication, VAPP and VDPV caused by OPV should be eliminated [44]. IPV provides protection against all three serotypes of poliovirus, which facilitates maintaining polio-free status [45]. Therefore, we could not ignore the possible social and economic impact of VAPP, such as vaccine hesitancy in Nigeria in 2003 [46], though the four-IPV schedule was cost-ineffective in our study. In Shanghai in 2019, in the context of implementing the free two-IPV-two-bOPV schedule, 47% of children’s parents chose the four-IPV schedule at their own expense or an IPV-containing combined vaccine instead (unpublished data), suggesting a high intent to receive IPV. Therefore, it indicated the feasibility of including the four-IPV immunization schedule into local EPI in resource-rich Chinese settings, such as metropolitan areas like Shanghai.

There are some limitations in this study. Cost of cold-chain services, human resources, and administration cost of human resources and administration of VAPP surveillance, treatment, follow-up, and compensation is difficult to accurately estimate. Thus, we hypothesized that they remain identical across the three polio immunization schedules. However, this study has made better estimations based on previous findings. In addition, cost of VAPP cases in this study was estimated only by the amount of compensation. Economic and social burden caused by VAPP was not included.

## 5. Conclusions

This study found that the two-IPV-two-bOPV and the four-IPV immunization schedules are currently cost-ineffective, which might be attributable to the difference in the cost between IPV and OPV and rare VAPP cases. However, they may be cost-effective by decreasing prices of IPV. Considering the risk of VAPP in the context of the four-tOPV schedule, IPV-containing schedules may be a feasible strategy to accelerate polio eradication efforts in China. In addition, it warrants further evaluation of the long-term effectiveness and benefits of IPV-containing schedules in China.

## Figures and Tables

**Figure 1 vaccines-09-01062-f001:**
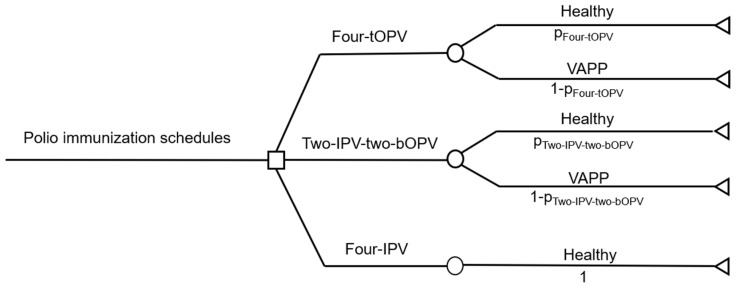
A decision tree model for cost-effectiveness analysis of three polio immunization schedules. The decision tree model had the following hypotheses: (1) 100% coverage of polio immunization in the study population; (2) no self-paid polio-containing vaccines in the study population; (3) no occurrence of VAPP after four-IPV immunization in the study population; (4) cost of cold-chain services, human resources, and administration remain identical across three polio immunization schedules. IPV, inactivated polio vaccine; VAPP, vaccine-associated paralytic poliomyelitis.

**Figure 2 vaccines-09-01062-f002:**
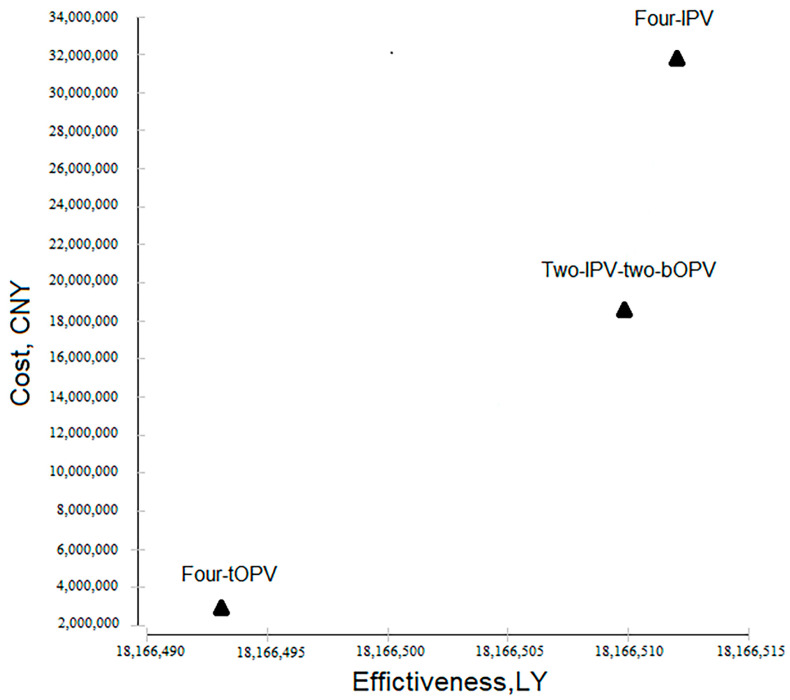
Cost-effectiveness analysis by using a decision tree model. Total cost of three polio immunization schedules in 2016 was used as the cost, and the sum of LY of local resident births in 2016 was used as the effectiveness. The four-tOPV immunization schedule was cost-effective; in contrast, both the two-IPV-two-bOPV schedule and the four-IPV schedule were cost-ineffective. bOPV, bivalent live attenuated oral polio vaccine; tOPV, trivalent live attenuated oral polio vaccine; IPV, inactivated polio vaccine; LY, life year.

**Figure 3 vaccines-09-01062-f003:**
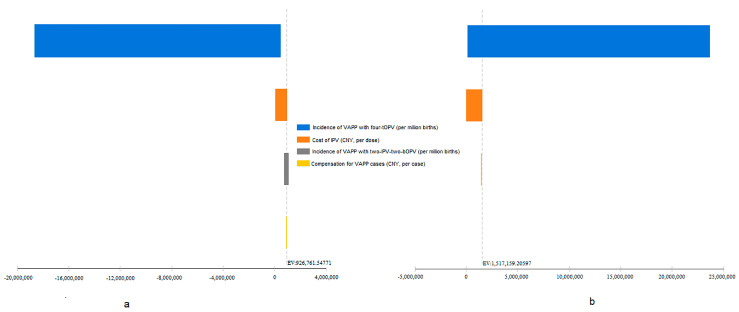
Tornado plot of the two-IPV-two-bOPV (**a**) and the four-IPV (**b**) immunization schedules, compared to the four-tOPV schedule, using univariate sensitivity analysis. The *X*-axis represented ICER_LY_. The incidence of VAPP cases with the four-tOPV schedule, incidence of VAPP cases with the two-IPV-two-bOPV schedule, compensation cost of VAPP cases, and cost of IPV were included in the analysis according to the range of values (Table 1). Positive and negative associations were further determined by the estimation of ICER, which were not presented in the figure. bOPV, bivalent live attenuated oral polio vaccine; tOPV, trivalent live attenuated oral polio vaccine; ICER, incremental cost-effectiveness ratio; IPV, inactivated polio vaccine; LY, life year; VAPP, vaccine-associated paralytic poliomyelitis.

**Figure 4 vaccines-09-01062-f004:**
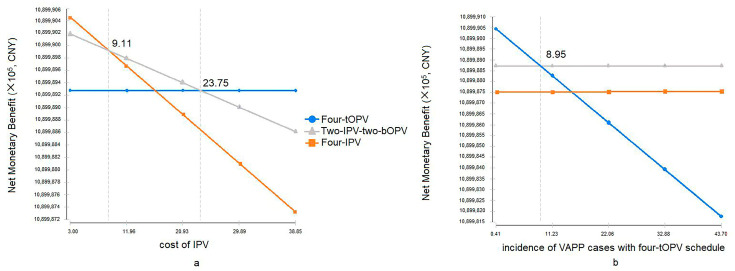
One-way sensitivity analysis net benefits graph for three polio immunization schedules. In (**a**), NMB of IPV-containing schedules was estimated over cost of IPV (*X*-axis), while that of the four-tOPV schedule was constant. In (**b**), NMB of the four-tOPV schedule was estimated over the incidence of VAPP cases with the four-tOPV, and those of IPV-containing schedules remained stable. The WTP in the decision tree model was set as CNY 600,000. NMB = total benefit – total cost, in which total benefit = effectiveness (sum of LY per year) × bound of ICERLY (that is WTP) and total cost was the total annual cost of different polio immunization schedules. bOPV, bivalent live attenuated oral polio vaccine; tOPV, trivalent live attenuated oral polio vaccine; IPV, inactivated polio vaccine; NMB, net monetary benefit; VAPP, vaccine-associated paralytic poliomyelitis.

**Table 1 vaccines-09-01062-t001:** Parameters for estimation of cost and effectiveness in the decision tree model.

Parameters	Baseline Value	Minimum Value	Maximum Value	Source of Data
Cost				
Government procurement price of vaccines (CNY/vial)				
tOPV (for 10-dose)	9.70	9.70	9.70	China Government Procurement Network
bOPV (for 10-dose)	27.20	27.20	27.20	China Government Procurement Network
sIPV (for 1-dose)	35.00	35.00	35.00	China Government Procurement Network
wIPV (for 1-dose)	39.00	39.00	39.00	China Government Procurement Network
Coefficient of loss				
10-dose per vial	1.80	1.80	2.50	Practical Regulations for Vaccination in Shanghai (2017 edition) [33]; SCDC
1-dose per vial	1.00	1.00	1.05	Practical Regulations for Vaccination in Shanghai (2017 edition) [33]; SCDC
Procurement ratio of sIPV:wIPV	2:1	1:1	2:1	SCDC
Procurement price of the low-temperature refrigerator (CNY)	15,000	10,000	20,000	SCDC
Depreciation of low-temperature refrigerator (years)	9	8	10	SCDC
Compensation for VAPP cases (CNY, per case)	748,005	748,005	1,250,000	SCDC; Shanghai Compensation Measures for Vaccination Related Adverse Events (trial edition) [35]; Yang LS [21]
Effectiveness				
Number of local resident births in Shanghai in 2016 (per 10,000)	21.84	21.84	21.84	Shanghai Statistical Yearbook 2017 [29]
Incidence of VAPP (per 1 million births)				
Four-tOPV schedule	6.20	0.41	43.70	SCDC; WHO study [39]
Two-IPV-two-bOPV schedule	0.70	0.00	1.56	SCDC; WHO position paper [1]; Sutter et al. [40]; Ivanova et al. [41]
Four-IPV schedule	0	0	0	
DALY caused by VAPP	14	13	14	Duintjer et al. [12], and Thompson et al. [14]

bOPV, bivalent live attenuated oral polio vaccine; tOPV, trivalent live attenuated oral polio vaccine; CNY, Chinese Yuan; DALY, disability-adjusted life year; ICER, incremental cost-effectiveness ratio; IPV, inactivated polio vaccine; SCDC, Shanghai Municipal Center for Disease Control and Prevention; VAPP, vaccine-associated paralytic poliomyelitis.

**Table 2 vaccines-09-01062-t002:** Total cost and effectiveness of three polio immunization schedules.

	Four-tOPV Schedule	Two-IPV-Two-bOPV Schedule	Four-IPV Schedule
Total cost (CNY 10,000 per year)	297.98	1856.42	3174.08
Cost of vaccines (CNY 10,000 per year)	152.53	1800.90	3174.08
Maintenance of low-temperature refrigerator (CNY 10,000 per year)	44.17	44.17	0
Compensation for VAPP cases (CNY 10,000 per year)	101.29	11.36	0
Incremental cost (CNY 10,000 per year)	reference	1558.44	2876.10
Number of VAPP cases prevented (per year)	reference	1.20	1.35
DALY averted (per year)	reference	16.83	18.96
ICER_VAPP_ (CNY 10,000 per VAPP case)	reference	1296.26	2124.02
ICER_DALY_ (CNY 10,000 per DALY)	reference	92.59	151.72

bOPV, bivalent live attenuated oral polio vaccine; tOPV, trivalent live attenuated oral polio vaccine; CNY, Chinese Yuan; DALY, disability-adjusted life year; ICER, incremental cost-effectiveness ratio; IPV, inactivated polio vaccine; VAPP, vaccine-associated paralytic poliomyelitis.

## Data Availability

The datasets generated during the current study are not publicly available due to privacy but are available from the corresponding author Yihan Lu on reasonable request.

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
