# Peer review of "Cost-Effectiveness of Three Poliovirus Immunization Schedules in Shanghai, China"

_vaccines, 2021, doi:10.3390/vaccines9101062_

Round 1

Reviewer 1 Report

This study explores the cost effectiveness of three polio immunization schedules in Shanghai, China using a decision tree model. The work is quite interesting, clear, and well-organized. The introduction provides enough background. The methods are described in detail. The Results and Discussion are correctly described. I only have some minor observations that are listed in the following lines.

  1. There are some typos. The work would benefit from close editing.
  2. Are all the schedules mentioned in this study currently in use in Shanghai? Please clarify.
  3. Table 2: what does reference indicate?
  4. Line 293: are these numbers per year?
  5. Add a list of abbreviations.

Author Response

Reviewer 1

  1. There are some typos. The work would benefit from close editing.

Authors’ response: Thank you for your detailed comments. We have corrected them in the manuscript.

  1. Are all the schedules mentioned in this study currently in use in Shanghai? Please clarify.

Authors’ response: Thank you for your comments. The two-IPV-two-bOPV schedule has replaced the traditional four-tOPV schedule in May 2016; furthermore, four-IPV schedule has replaced the two-IPV-two-bOPV schedule since October 2020. Thus, during this study, the four-IPV schedule is the single polio immunization program in Shanghai. We have revised accordingly for more clarification in the Materials and Methods (Lines 80-83).

  1. Table 2: what does reference indicate?

Authors’ response: Thank you for your comments. In this study, we determined the Incremental Cost-Effectiveness Ratio (ICER) to compare the difference in the cost-effectiveness across the three polio immunization schedules, which warrants a reference group. Thus, we defined the four-tOPV schedule as a reference group, and subsequently calculated incremental cost (CNY 10,000 per year), number of VAPP cases prevented (per year), DALY averted (per year), ICERVAPP (CNY 10,000 per VAPP case), and ICERDALY (CNY 10,000 per DALY) of the two IPV-containing schedules.

  1. Line 293: are these numbers per year?

Authors’ response: Thank you for your comments. Yes, these are annual numbers.

  1. Add a list of abbreviations.

Authors’ response: Thank you for your comments. We have added it in the end of the manuscript (Lines 532-536).

Reviewer 2 Report

This is very interesting study on three poliovirus immunization and cost effective module. This evaluation might be useful for the researchers working on different vaccines. I have some minor comments-

  1. Abstract should me more focused. Central message of this study is missing.
  2. Most of the references are quite old in the introduction section. Authors should update the introduction section with recent findings and citations.
  3. Figures are cited cited in the manuscript text. This should be cited properly with better description. Authors should discuss properly in the discussion section also.
  4. Authors should add the conclusion section separately with possible future perspectives of this studies.    

Author Response

Reviewer 2

  1. Abstract should me more focused. Central message of this study is missing.

Authors’ response: Thank you for your valuable comments. We have revised accordingly in the Abstract.

  1. Most of the references are quite old in the introduction section. Authors should update the introduction section with recent findings and citations.

Authors’ response: Thank you for your suggestion. We have added 5 references published during 2014-2017, and removed 2 references published in 1984 and 2009.

  1. Figures are cited in the manuscript text. This should be cited properly with better description. Authors should discuss properly in the discussion section also.

Authors’ response: We appreciate your valuable suggestion. We have revised and elaborated more in the text and figure legends for the figures.

  1. Authors should add the conclusion section separately with possible future perspectives of this studies.

Authors’ response: Thank you for your comments. We have added a Conclusions section following the Discussion.
